# SiC Nanoparticles Strengthened Alumina Ceramics Prepared by Extrusion Printing

**DOI:** 10.3390/ma16062483

**Published:** 2023-03-21

**Authors:** Jian Wu, Hai Zheng, Mingliang Tang, Zhuqing Yu, Zhigang Pan

**Affiliations:** 1College of Materials Science and Engineering, Nanjing Tech University, Nanjing 211816, China; 2Donghai Institute of Advanced Silicon-Based Materials, Nanjing Tech University, Nanjing 222300, China

**Keywords:** 3D printing, SiC nanoparticles, alumina, mechanical properties

## Abstract

Extrusion-free-form printing of alumina ceramics has the advantages of low cost, short cycle time, and high customization. However, some problems exist, such as the low solid content of ceramic paste and the unsatisfactory mechanical properties of pure alumina ceramics. In this study, SiC nanoparticles were used as a reinforcement phase added to the alumina ceramic matrix. Methylcellulose is used as the binder in the raw material system. Ammonium polyacrylate is used as a dispersant to change the rheological properties of the slurry and increase the solid content of ceramics. SiC nanoparticle-strengthened alumina ceramics were successfully prepared by the extrusion process. The relative settling height and viscosity of ceramic slurries were characterized. The sintering shrinkage of composite ceramics was tested. The flexural strength, fracture toughness, and hardness of the ceramics were characterized. The strengthening and toughening mechanisms of the composite ceramics were further explained by microscopic morphology analysis. Experimental results show that when the content of the dispersant is 1 wt.%, the rheological properties of the slurry are the best. Maximum measured bending strength (227 MPa) and fracture toughness (5.35 MPa·m^1/2^) were reached by adding 8 wt% SiC nanoparticles; compared with pure alumina ceramics, flexural strength and fracture toughness increased by 42% and 41%, respectively. This study provides a low-cost and effective method for preparing ceramic composite parts.

## 1. Introduction

Alumina ceramics are characterized by high hardness, high temperature, corrosion, and good wear resistance and are widely used in ceramic tools, ceramic bearings, aviation blades, medical instruments, and other industries [1,2,3,4]. However, the alumina ceramics’ brittleness and low fracture toughness limit their applications [5,6,7]. Traditional ceramic preparation processes such as injection molding, dry pressing, and isostatic pressing [8,9,10] cannot prepare ceramic parts with complex shapes and high precision, requiring custom molds and high preparation costs.

The fracture toughness of alumina ceramics can be increased by adding a reinforcing phase to the alumina matrix. Introducing 3D printing technology to prepare alumina ceramics can reduce the cost and preparation cycle and be customized to print complex-shaped parts [11]. Researchers added second-phase materials such as particles, fibers, and whiskers [12,13,14] to the alumina matrix to change the intergranular fracture mode of alumina ceramics and enhance the fracture toughness and strength of alumina ceramics.

Yin et al. [15]. Developed a hot-pressing technique to fabricate TiC particle-reinforced Al_2_O_3_-based composites using cobalt as a sintering additive. Experimental results have shown that the composites containing 6 vol.% nanoscale TiC have the best overall mechanical properties. The maximum fracture toughness was 8.3 MPa·m^1/2^. Dong et al. [16]. Developed a hot-pressing technique to fabricate nano-/micro-sized SiC/Al_2_O_3_ composites. The composites had the highest fracture toughness of 7.6 MPa·m^1/2^ when the silicon carbide content was 5 wt.%. The composites had the highest bending strength when the silicon carbide content was 20 wt.%. The intragranular fracture was observed in the composites. The increase in fracture toughness was mainly attributed to crack deflection.

Compared with the traditional ceramic forming method, the preparation of alumina ceramics by 3D printing technology has the advantages of fast forming and low cost. Ceramic parts with complex geometric shapes can be prepared according to 3D models without using molds [17,18,19]. The leading ceramic 3D printing technologies are fused deposition modeling (FDM) [20], stereolithography (SLA) [21], digital light processing (DLP) [22], selective laser sintering (SLS) [23], extrusion freeform technology (EFF) [24], etc. Unlike other techniques for preparing alumina ceramics that require a laser or high-cost resin feedstock [25,26,27,28], extrusion free-form technology is an additive manufacturing technique for building 3D objects through layered paste deposition of ceramics by material extrusion layer by layer. The process has the advantages of high molding efficiency, broad applicability, and low cost. However, the disadvantage is the low accuracy and strength of printed parts. This paper reports a method of preparing composite ceramics by extrusion printing based on the slurry form. A new raw material system using methylcellulose as a binder and polyethylene glycol as a plasticizer has improved the stability of the slurry and produced a ceramic slurry with high solid phase content. Using ammonium polyacrylate as a dispersant reduced the viscosity and improved the rheological properties of the slurry. Ultrasonic dispersion and ball milling improved the dispersion of the SiC nanoparticles in the alumina matrix. The effects of dispersant and SiC nanoparticle content on the rheological properties of the slurry were investigated. The microstructure of the ceramic body was characterized by SEM, and the toughening mechanism of composite ceramics was analyzed by ceramic fracture morphology. The mechanical properties of composite ceramics, such as Vickers hardness, flexural strength, and fracture toughness, were tested and compared with those of pure alumina ceramics. The effects of SiC nanoparticle content and sintering temperature on the mechanical properties and microstructure of ceramic samples were investigated.

## 2. Materials and Methods

### 2.1. Materials

α-alumina powder (purity 99.9%, average particle size 2 μm) (Guangdong Dongguan New Material Technology Co., Ltd., Dongguan, China) was used as the matrix ceramic powder. SiC powder (99.9%, average particle size 50 nm). Shanghai Naiou Nano Technology Co., Ltd., Shanghai, China) was used as a reinforcing phase and MgO (99.9%, average particle size 1 micron) as a sintering aid to improve sintering activity and reduce sintering temperature. Ammonium polyacrylate was selected as a dispersant, polyethylene glycol as a plasticizer, and methyl cellulose as a binder. Deionized water was used as the solution.

### 2.2. Preparation of Ceramic Slurry

Polyethylene glycol (8 wt.%) and methylcellulose (1 wt.%) were added to deionized water (90 wt.%) to prepare a premixed solution. Methylcellulose is used as a binder to maintain the shape of the extruded paste. Then dispersant ammonium (1 wt.%) polyacrylate was added to increase slurry fluidity and avoid agglomeration of particles. A premixed solution is mixed in a vacuum homogenizer (ZYMC-350VS, Shenzhen ZYE Science & Technology Co., Ltd., Shenzhen, China). SiC powder (99.9% purity, average particle size 50 nm, Shanghai Naio Nanotechnology, Shanghai, China) was added to the premixed solution and dispersed using ultrasound. A composite powder composed of α-Al_2_O_3_ powder (99.9% purity, average particle size 2 mm, Guangdong Dongguan New Material Technology, Dongguan, China) and MgO (0.8 wt.%, 99.9% purity, average particle size 1 mm) was added into the premix solution. The SiC/Al_2_O_3_ mixture was obtained after 12 h of ball milling. The SiC nanoparticle content was 2%, 4%, 6%, 8%, and 10 wt.%, respectively. The slurry is further mixed using a vacuum homogenizer to eliminate air bubbles. The viscosity of the slurry is tested by a dynamic viscometer.

### 2.3. 3D printing of the Ceramic Green Body

First, the 3D model (Figure 1) is sliced in the CURA software (5.1.0) package to generate a Gcode format file that can be recognized by the 3D printer.

Figure 1 shows the schematic diagram of 3D printing. The prepared slurry is poured into the mud storage tank. The gas pressure generated by the air compressor transports the slurry to the screw module. The screw motor drives the screw to rotate counterclockwise to transport the mud to the nozzle. The XYZ three-axis motor and the screw motor control the precise displacement of the nozzle according to the model parameters.

Ceramics are printed after setting the printing parameters. Print a few circles outside the green body to test the continuity of printing, which also helps the green body detach from the substrate after drying. The printed ceramic green bodies are shown in Figure 2.

The nozzle diameter is set to 1 mm. The layer height is set to 0.7 mm. The printing parameters are shown in Table 1:

### 2.4. Debinding and Sintering

The printed ceramic green bodies were dried in an oven at 60 °C for 12 h. The dried samples were put into a tube furnace for debinding and sintering under an argon atmosphere. The temperature profile for debinding and sintering is shown in Figure 3.

The process of debinding and sintering is divided into three steps. The first is low-temperature debinding. The samples were heated to 200 °C at a rate of 1 °C/min to remove moisture and some polyethylene glycol from the green bodies. then heated to 420 °C at a heating rate of 2 °C/min and maintained at 420 °C for 1 h. Polyethylene glycol, ammonium polyacrylate, and methylcellulose were removed. Finally, samples are heated to 800 °C at a rate of 3 °C/min. Then, keep the temperature at that level for 2 h to completely remove the additives. The temperature was raised to 1400 °C, 1500 °C, and 1600 °C at a heating rate of 5 °C/min for high-temperature sintering and then kept at 1600 °C for 2 h. Finally, the temperature was lowered to room temperature at a cooling rate of 5 °C/min.

### 2.5. Characterization

The viscosity of the slurry with different dispersant contents was tested with a viscometer to evaluate the rheological properties of the slurry. The density and open porosity of sintered samples were determined by the Archimedes method.

The ceramic slurry is poured into a graduated cylinder for settling experiments. Relative sedimentation height (RSH) was calculated as follows:(1)$$R=\frac{H_{1}}{H}\times 100\%$$
where $H_{1}$ is the height of the slurry sedimentation layer (mm), and H is the total height of the slurry (mm).

The length of samples before and after the sintering process was measured by a Vernier caliper, and the shrinkage of ceramic samples was obtained. The formula is as follows:(2)$$L_{c}=\frac{L_{0}-L}{L_{0}}\times 100\%$$
where $L_{0}$ is the size of the samples before sintering(mm), L the size of the samples after sintering (mm).

The flexural strength and fracture toughness of the sintered samples were measured using a universal testing machine. The standard size of the sample is 36 mm × 4 mm × 3 mm. The span is 30 mm, and the loading rate is 0.5 mm/min. The strength value is taken as the average of 10 specimens. The flexural strength of the samples was measured by the three-point bending method. The formula is as follows:(3)$$\sigma f=\frac{3FL}{2bh^{2}}$$
where σf is the flexural strength (MPa), *F* the maximum load (N), *L* the span (30 mm), *b* the width of the sintered sample (mm), and *h* the height of the sintered sample (mm).

The fracture toughness of sintered samples was tested by the SENB method. Samples are cut and ground to standard sizes. The sample size is 30 mm × 4 mm × 2 mm, the notch width is 0.25 mm, and the notch depth is 1.7 mm. The Vickers hardness of the samples was tested by a Vickers hardness tester. The microstructure and fracture morphology of ceramic parts were detected by scanning electron microscopy, and the toughening mechanism of composite ceramics was analyzed.

## 3. Results and Discussion

### 3.1. Rheological Properties of Ceramic Slurry

In the extrusion printing process, dispersants are added to the slurry to prevent precipitation and solvent migration from occurring and to avoid agglomeration of the SiC nanoparticles. A stable slurry ensures the continuity of extrusion and the acquisition of raw ceramic blanks with a homogeneous internal structure. As a result, the strength of the ceramic material can be increased. The dispersion and stability of ceramic slurries were evaluated by their relative sedimentation height. Figure 4 shows the relationship between the relative sedimentation height of the ceramic slurry and the dispersant content. As expected, the sedimentation height of the slurry with good dispersion is higher. Ammonium polyacrylate can improve the dispersion of ceramic slurry. When no dispersant was added, the fastest sedimentation rate was observed. The relative sedimentation height was 45%, and the stability of the slurry was low. Excessive dispersant will produce multi-layer adsorption in the slurry and reduce the dispersion of the slurry.

Furthermore, the high viscosity of the polymer itself is easy to bridge, resulting in a slight decrease in the fluidity of the slurry and an increase in viscosity. The experimental results showed that the highest sedimentation height and slowest settling rate were obtained for slurries containing 1 wt.% dispersant, which was most effective in reducing the sedimentation of ceramic slurries to obtain stable slurries.

When the optimal content of the dispersant was determined, the effect of different contents of SiC nanoparticles on the rheological properties of the ceramic slurry was studied. The addition of SiC nanoparticles changes the viscosity of the paste. In order to ensure that extrusion printing can be carried out continuously, the viscosity of the slurry used for extrusion should be less than 10,000 mPa·s. Therefore, the content of SiC nanoparticles needs to be in an appropriate range to prevent excessive SiC nanoparticles from exceeding the viscosity range required for extrusion printing.

Figure 5 shows the effect of SiC nanoparticles with different solid phase contents on the viscosity of the slurry at a dispersant content of 1 wt.%. Overall, the addition of nanoparticles increases the slurry’s viscosity and shear stress. On the one hand, nanoparticles have a small particle size, a large specific surface area, and a high surface energy, which leads to spontaneous agglomeration. On the other hand, the increase in nanoparticle content increases the difficulty of its dispersion. It can be seen from Figure 5 that the slurry viscosity growth trend has a lower gradient when the SiC nanoparticle content is 0–8%. When the content is 10%, the viscosity starts to increase significantly. Even though the viscosity can be reduced by increasing the dispersant content, the stability of the slurry will be reduced. The printing will produce discontinuities, which will significantly impact the microstructure and mechanical properties of the raw ceramic green body. Therefore, the content of SiC nanoparticles should be at most 8% (Figure 5) for preparing composite ceramics by the extrusion molding process.

### 3.2. Physical Properties

Figure 6 shows the effect of the content of SiC nanoparticles on the sintering shrinkage of ceramic samples. Ceramic materials shrink and densify significantly during the sintering process, and the increase in densification results in enhanced mechanical properties. Sintering shrinkage is an important parameter to characterize the sintering density. It is observed that with the increase in SiC nanoparticle content, the shrinkage rate in the three directions increases at first and then decreases.

When the content of SiC nanoparticles is 0–4%, the shrinkage rate shows an increasing trend in the length, width, and height directions. The reason is that SiC nanoparticles fill the voids between alumina particles, replacing the original pores and other defective locations. This results in fewer internal defects and higher densities in ceramics. The ceramic shrinkage decreases when the SiC nanoparticle content exceeds 4%. Because in the same sintering system, when the content of SiC nanoparticles is low, the particles can be uniformly dispersed in the alumina ceramic matrix, so the density of the ceramic is higher. As the content of SiC nanoparticles increases, the viscosity of the slurry increases, and the fluidity of the slurry decreases. At the same time, the addition of excessive nanoparticles causes agglomeration, which affects the sintering densities and makes the shrinkage decrease instead. It can be found that the magnitude of shrinkage along the height direction is higher than in the other two directions. Because the extrusion-based printing process employs a layered molding approach, a buildup in the height direction occurs, resulting in varying degrees of shrinkage of the sintered sample. The gaps remaining between the extruded layers and coils also make the shrinkage in the Z direction different from that in the other two directions. The stability of the paste is low, and the underfilling of the paste also causes porosity between the extruded coils, resulting in higher shrinkage in the Z-direction. Gaps between the extrusion coils can be reduced by adjusting the print speed and increasing the nozzle extrusion flow.

Table 2 shows the relevant properties of the ceramic samples with different SiC nanoparticles and composite ceramic contents at a sintering temperature of 1600 °C. As shown in Table 2, the ceramic hardness increased from 15.58 GPa to 16.74 GPa as the SiC nanoparticle content increased from 0% to 4%. The relative density of ceramics first increased to 97.61%, and then began to decrease.

The maximal flexural strength and fracture toughness of ceramics are 227 ± 7.33 MPa and 5.35 ± 0.46 MPa·m^1/2^, respectively. Compared with pure Al_2_O_3_ ceramics, the mechanical properties of SiC nanoparticle-toughened Al_2_O_3_ ceramics were significantly improved, and the addition of silicon carbide nanoparticles increased the flexural strength by 41% and the fracture toughness by 42%.

The mechanical properties of composite ceramics with different silicon carbide contents are closely related to the microstructure of the ceramics. For pure alumina ceramics without SiC nanoparticles, there are many pores in the alumina matrix, leading to low flexural strength.

With the increase in SiC nanoparticle content, the porosity of the alumina matrix decreases. The reduction in the size and number of pores results in improved mechanical properties for the ceramic. The flexural strength and fracture toughness decrease when the SiC nanoparticle content exceeds 8%. The reason is that as silicon carbide content increases, excess nanoparticles agglomerate among the alumina matrix. The sintering activity of the ceramics is reduced, which affects the mechanical properties of the ceramics.

Furthermore, the viscosity curve in Figure 5 shows that increased SiC nanoparticle content leads to increased paste viscosity. The reduced fluidity of the paste affects the printing accuracy of the ceramic blank.

### 3.3. Effect of Temperature Regime on Mechanical Properties of Ceramics

The flexural strength and fracture toughness of the composite ceramics sintered at 1400 °C, 1500 °C, and 1600 °C were measured, as shown in Figure 7. It can be seen that as the temperature increases, the mechanical properties of the ceramics also increase. The flexural strength and fracture toughness are low when the sintering temperature is 1400 °C. From the microscopic morphology of the ceramics, it can be seen that the overall density of the composite ceramics is low and there are internal defects. Thus, the mechanical properties of the specimens are poor. The fracture toughness of ceramic materials is lower when sintered at lower temperatures. The inter-particle diffusion phenomenon is not apparent. The van der Waals force becomes the primary bonding mode between the particles. Therefore, the crack extension driving force is much larger than the inter-particle bonding force.

As the sintering temperature increases, the diffusion phenomenon between the particles becomes more and more apparent. The resulting liquid phase makes the ceramics more tightly bound, increasing interfacial energy. According to the SEM image (Figure 8c) shown at temperatures up to 1600 °C, the particles are more tightly bound, and the mechanical properties of the ceramics are significantly improved. It is also observed that the SiC nanoparticles have little effect on the mechanical properties of alumina ceramics at lower temperatures. Overall, the sintering regime dramatically influences the strength of composite ceramics.

### 3.4. Microstructure

Sintering increases the densification and mechanical strength of the ceramic green body, so the sintering temperature is a crucial parameter. Figure 8 shows the microscopic morphology of the green body and the sintered samples at 1400 °C and 1600 °C. With the increase in sintering temperature, the electron microscope scan results show that the grains gradually form grain boundaries. The increase in sintering temperature increases the densities of the ceramics, and the particles are more tightly bound. The green body (Figure 8a) has many pores (red circles) and loosely bound particles. After the green body was sintered at 1400 °C (Figure 8b), the pores gradually decreased, the ceramic densities increased, and particle size increased, but the uniformity was poor. Because the sintering temperature of alumina ceramics has yet to be reached and the sintering drive force is insufficient. The particles were tightly bound when sintered at 1600 °C (Figure 8c), producing a liquid phase with few pores observed. The gradual blurring of the grain boundaries of the material at high temperatures relative to 1400 °C is caused by the softening of the glass phase. Therefore, the sintering temperature is critical to ceramics’ microstructure and mechanical properties and is crucial in achieving ceramic densification.

Figure 9 shows the fracture morphology of composite ceramics with different SiC nanoparticle contents sintered at 1600 °C. Figure 9a shows the microscopic morphology of pure Al_2_O_3_ ceramics without adding silicon carbide. The grains are of different sizes and shapes and are abnormally large.

Increasing SiC nanoparticle content leads to grain refinement and uniform grain size distribution. Grain refinement and homogenization reduce ceramic defects and stress concentration and improve the fracture toughness of composite ceramics. As seen in Figure 9b, some of the nanoparticles enter the middle of the grains after adding nanoparticles, forming an intragranular structure. The fracture morphology of the ceramic samples also shows holes on the fracture surface after grain extraction, which indicates that the grain extraction also improves the fracture toughness of the ceramics.

In addition to grain refinement and intergranular fracture, microcracking toughening mechanisms also play an important role in the strength enhancement of ceramics. Figure 10b. Due to nanoparticles within the matrix, many microcracks are generated when subjected to residual stress expansion, further refining the grain. When the content of SiC nanoparticles is increased to 10%, microcracks appear at the grain boundaries, and stress concentration occurs when pressure is applied, reducing the mechanical properties of the ceramic.

It can be seen from Figure 9c that agglomerations of nanoparticles existed within the matrix. It produces a non-uniform microstructure, reducing the ceramic’s sintering density and mechanical properties. The primary fracture mode of ceramics is a mixture of intergranular and transgranular fractures, with intergranular fractures being the primary type. The SiC nanoparticles distributed between the alumina crystals are affected by residual stress, have an effect at the interface, and increase interfacial strength [29,30,31]. At the same time, the crack deflection generated by the matrix dissipates a large amount of energy. It improves the mechanical properties of composite ceramics.

## 4. Conclusions

A new slurry extrusion-based 3D printing process was used to prepare SiC nanoparticle-strengthened alumina ceramics. Ammonium polyacrylate was used as the dispersant and methylcellulose as a binder. Composite ceramic pastes with different contents of SiC nanoparticles were prepared. The raw silicon carbide-reinforced alumina was prepared by extrusion 3D printing, and the sintered composite ceramic samples were obtained after heat treatment. The influence of the raw material system on the printing process was investigated. The effects of SiC nanoparticle content and sintering temperature on the relative density, hardness, flexural strength, fracture toughness, and microscopic morphological structure of ceramic samples were studied. The results are as follows:

The best rheological performance and stability of the slurry are achieved when the content of ammonium polyacrylate is 1 wt.%. Overall, the SiC nanoparticle content increases the viscosity of the slurry. When SiC nanoparticle content is 10 wt.%, the slurry’s viscosity increases sharply, the printing appears discontinuous, and the slurry clogs the nozzle.

The mechanical properties of the ceramics are superior when the sintering temperature is 1600 °C. When the sintering temperature is 1600 °C, the flexural strength and fracture toughness of pure alumina ceramics are 131 ± 4.26 MPa and 3.76 ± 0.16 MPa·m^1/2^, respectively. As the content of SiC nanoparticles increases, the flexural strength and fracture toughness also increase. The best fracture toughness of the composite ceramics was achieved when the SiC nanoparticle content was 8 wt.%, reaching 5.35 ± 0.46 MPa·m^1/2^. Fracture toughness is increased by 44% compared to pure alumina ceramics. The fracture toughness decreases when the SiC nanoparticle content exceeds 10 wt.%.

The SEM images show that the composite ceramics have a high density when sintered at 1600 °C. The fracture morphology shows that the fracture mode is a mixture of transgranular and intergranular fractures, with intergranular fracture being the primary mode. The addition of SiC nanoparticles inhibits abnormal alumina grain growth, refines grains, and improves fracture toughness.

## Figures and Tables

**Figure 1 materials-16-02483-f001:**
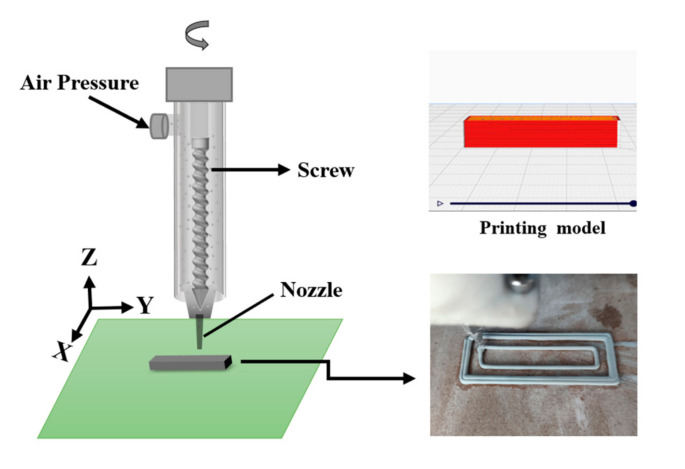
Schematic diagram of the 3D printing process for ceramic parts.

**Figure 2 materials-16-02483-f002:**
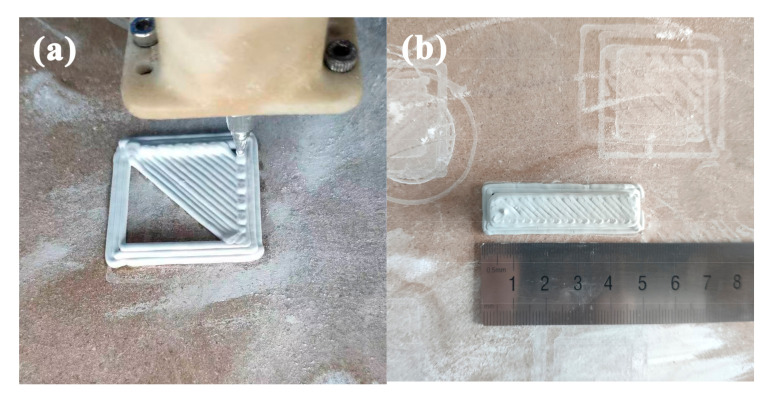
Extrusion printing of ceramic green bodies. (**a**) Ceramic green body being printed; (**b**) Printed ceramic green body.

**Figure 3 materials-16-02483-f003:**
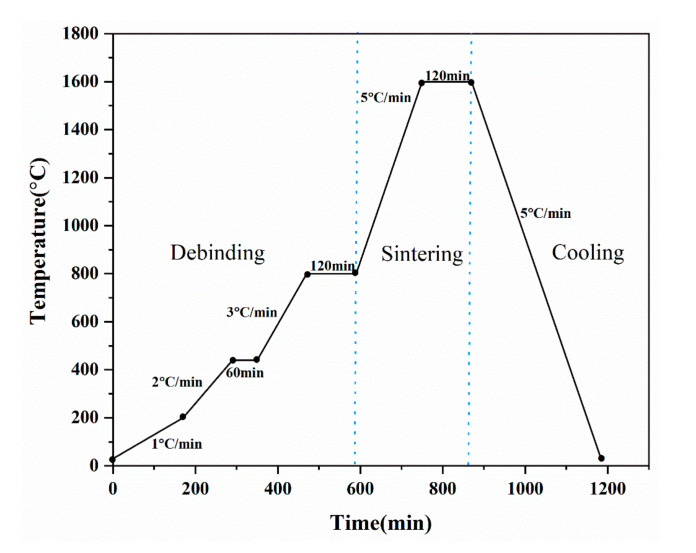
Debinding and sintering curves of composite ceramics.

**Figure 4 materials-16-02483-f004:**
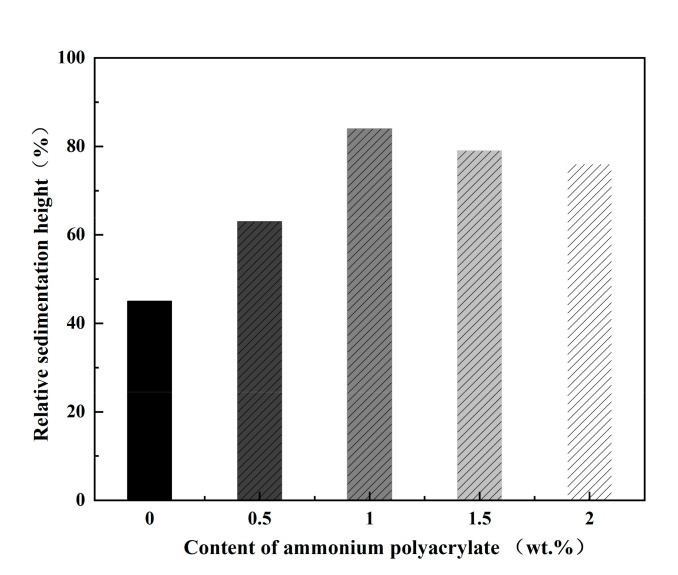
Relative sedimentation height of ceramic slurry.

**Figure 5 materials-16-02483-f005:**
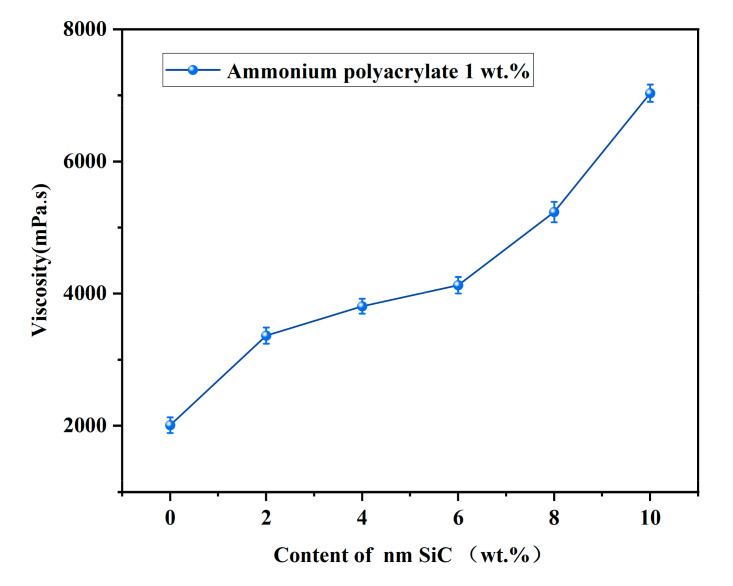
The viscosity of the slurry at different SiC nanoparticle contents.

**Figure 6 materials-16-02483-f006:**
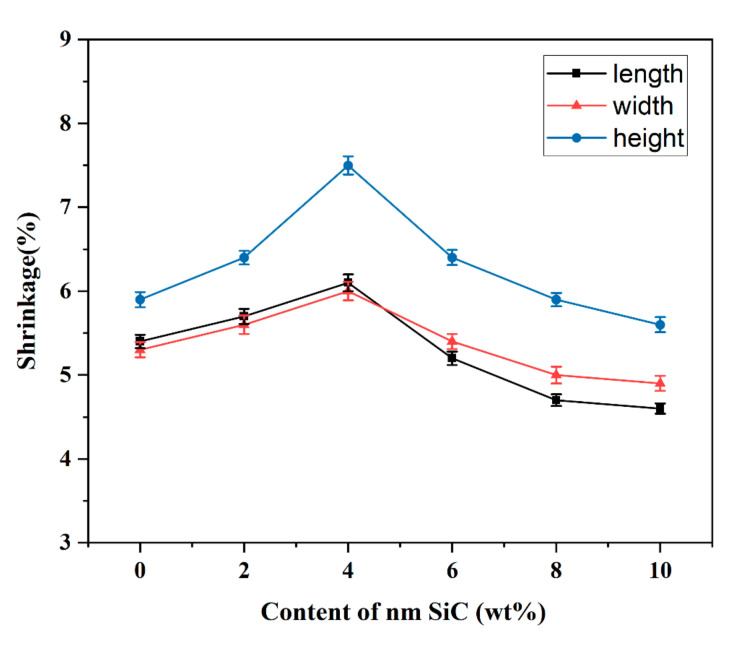
Shrinkage of the sintered samples in the length-width-height directions.

**Figure 7 materials-16-02483-f007:**
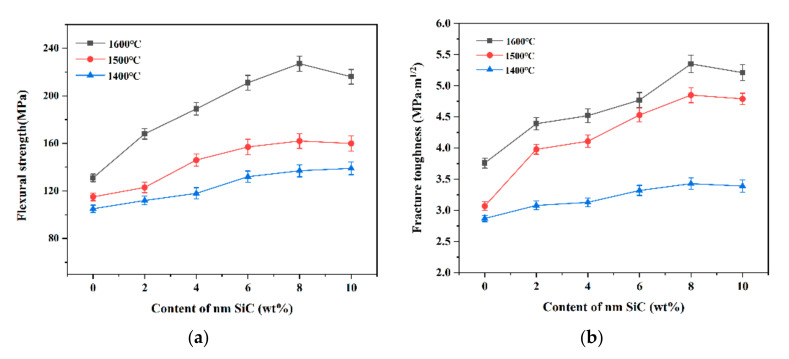
Mechanical properties of composite ceramics with different contents of silicon carbide at different temperatures. (**a**) flexural strength; (**b**) fracture toughness.

**Figure 8 materials-16-02483-f008:**
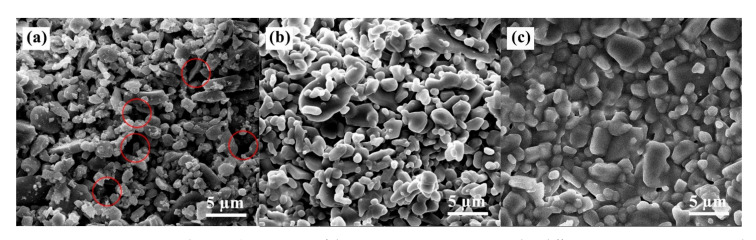
SEM images of the composite ceramics sintered at different temperatures. (**a**) green body; (**b**) 1400 °C; (**c**) 1600 °C.

**Figure 9 materials-16-02483-f009:**
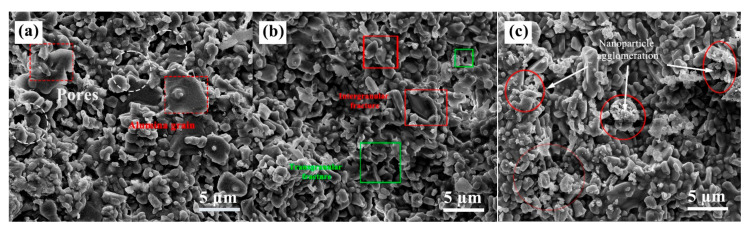
SEM images of sintered ceramics prepared with different SiC contents. (**a**) 0 wt%; (**b**) 8 wt%; (**c**) 10 wt%.

**Figure 10 materials-16-02483-f010:**
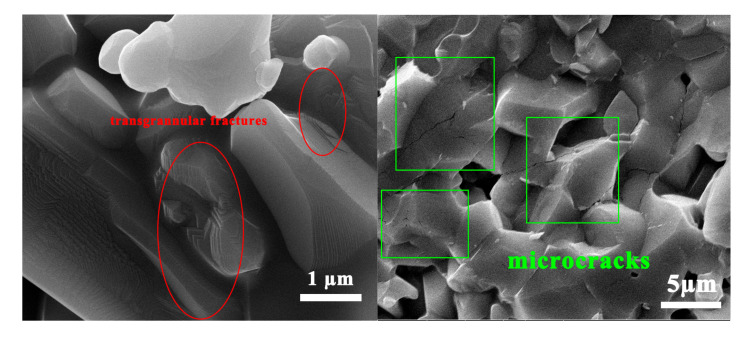
Micrographs of Fracture Toughness Enhancement Mechanism Analysis. (**a**) transgranular fracture. (**b**) microcrack toughening.

**Table 1 materials-16-02483-t001:** Printing parameters of EFF.

**Printing Parameter**	**Value**
Screw rotating speed	30 r/min
Printing speed	20 mm/s
Layer height	0.7 mm
Nozzle diameter	1 mm

**Table 2 materials-16-02483-t002:** Mechanical properties of ceramic samples sintered at 1600 °C.

**SiC** **Content**	**Vickers Hardness (%)GPa**	**Flexural Strength (MPa)**	**Fracture Toughness(%) MPa·m^1/2^**	**Total Porosity (%)**	**Bulk Density (g/cm^3^)**
0%	15.58 ± 0.49	131 ± 4.26	3.76 ± 0.16	28	3.36
2%	16.21 ± 0.46	168 ± 5.37	4.39 ± 0.23	24	3.41
4%	16.74 ± 0.51	189 ± 6.23	4.52 ± 0.28	20	3.42
6%	15.67 ± 0.48	211 ± 7.15	4.77 ± 0.36	19	3.42
8%	15.53 ± 0.45	227 ± 7.33	5.35 ± 0.46	16	3.47
10%	15.45 ± 0.46	216 ± 7.21	5.21 ± 0.34	17	3.43

## Data Availability

Not applicable.

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
