# Peer review of "SiC Nanoparticles Strengthened Alumina Ceramics Prepared by Extrusion Printing"

_materials, 2023, doi:10.3390/ma16062483_

Round 1
Reviewer 1 Report
An interesting paper but a number of issues must be corrected before acceptance. In particular:
1) English usage is often unclear or ambiguous or syntactically incorrect (e.g. line 42: "Yin et al [15]. Prepared micro-scale..., or line 171: "...affects the dispersion effect of the slurry") - please ask a native Engish speaker to check the text.
2) Section 2.2. Please give composition of PEG+MC+water solution. Also amount of Amm. polyAcr. used and how was it mixed in (Ultrasonic?). Also, amount of MgO used in alumina. What kind of viscometer was used (dynamic or kinetic?).
3) Line 114: What is meant by "...best printing effect.."? What criteria did you use to ascertain this?
4) Section 2.2. What was the strength of the un-dried green bodies? Can this technique be used for self-supporting 3D structures? It is mentioned that the nozzle diameter is 1mm but Figure 2 shows a sample with much thicker extruded coils. Please explain. How did you eliminate inter-coil gaps and interfaces between the extruded coils?
5) Lines 125 onwards: please explain the reasons for choosing these temperatures for debinding (200C, 420C, 800C) and what happens at each stage. Please specify that you sintered at 3 temperatures: 1400, 1500 and 1600C
6) Section 2.4. How did you shape the specimens after sintering to obtain a uniform shape in order to characterise them? By grinding? What about the interfaces and gaps between the extruded coils? Did you look for any inter-coil gaps after sintering?
For toughness, what was the SENB notch depth and the specimen size? What hardness (Vickers) load? How many specimens were tested in each test?
7) Line 167: "Polyammonium methacrylate"?? or "Ammonium polyacrylate" as mentioned in 2.2, or "Ammonium polymethacrylate" mentioned later? Be consistent!
8) Line 184, "MPa.s" should be "mPa.s"
9) Line 195: ".. is relatively flat.." is misleading - better write "...has a lower gradient.."
10) Lines 220 onwards. The explanation of the sharp decrease in shrinkage as SiC content increase after 4% in fig. 6 is not reflected in Figure 5 where even 6% seems to give only slight increase in viscosity. Also, the much larger shrinkage in the Z direction is not explained well. I think it must have something to do with the gaps remaining between the extruded layers and coils. Interestingly, the hardness follows the shrinkage and density results, as would be expected.
11) The maximum of strength and toughness at about 8% SiC is very surprising since the density is much lower than at 0-4%. Please give the variation in density results
The very small spread in the strength and toughness results for such materials is also very surprising. I assume these are average values, but what are the variations? StdDev? Please give the number of specimens tested and the spread of results.
12) Line 244. Fig5a does not exist. Did you mean 8a?
13) Lines 246 onwards: "...the nanoparticles fill the pores in the 246 alumina matrix..". This is not correct - in Figure 9b, there are still many pores in the structure, which reflects correctly the low density reported.
14) Line 281: "Sintering increases bulk density.." - please show the density results and compare with hardness, strength and toughness. The results you show are surprising.
15) Fig 8: Are these photos of outer surface or of fracture surface? How much SiC in these specimens?
16) Line 304: "..Hinder the diffusion of alumina grain boundaries." How did you conclude this? Did you see many SiC grains along grain boundaries?
17) Line 315 and 325 and 356: I cannot see any transgrannular fractures in these photos. Do you have higher magnification/resolution photos? From what I can see, the fracture surface shows only intergrannular features.
18) Line 317. Did you see any clear microcracks? Can you show them?
In general, the conclusions are not supported by your results. There is a clear discrepancy between low density and maxima of strength and toughness. Please check your results again.
Reviewer 2 Report
Reviewer Comment
Manuscript Number: Materials-2290747
Dear Editor,
1-Line 12: sic should be corrected (SiC).
2-wt.% should replace wt% throughout the submitted article.
similarly, vol. % instead of vol%
3-section 2.2: Line 95, are these ratios in wt.% or vol.%? the authors should reveal it.
4-in Figure 3, the authors described the debinding process. However, they gave no reason for this distribution.
5-line 154-155, for fracture toughness measurement, the acronym SENB can be used.
6-Line 274: which SEM micrographs authors does mean? Figure 8?
It should be mentioned by the authors.
7-Why the authors did not use the temperature of 1700 C to investigate the higher temperature effect on the mechanical properties and density of Al2O3-SiC ceramics? As it is clear from the results, the higher the sintering temperature, the better mechanical properties.
8- transgranular fracture ...should replace the capital starting letter throughout this manuscript.
9-the paragraph, line 322-329 is true. however, it should be supported by references.
10-chemical descriptions in the reference should be corrected. Al2O3, Al2O3, ZrO2, ZrO2, ...etc.
9-
Round 2
Reviewer 1 Report
Thanks for revising and improving the manuscript, however please make these additional corrections:
1) In Table 2, please add the total porosity as well (%) to show the effect of addition of SiC. It is important since it is used in your discussions of your toughness and strength results.
2) From your SEM photos it seems obvious that intergrannular fracture is predominant to transgrannular fracture, as expected for such microstructure and such high levels of porosity. Please correct this in your discussion and conclusions otherwise it looks contradictory.
